# A Human-Centered Design Approach to Develop Oral Health Nursing Interventions in Patients with a Psychotic Disorder

**DOI:** 10.3390/ijerph20043475

**Published:** 2023-02-16

**Authors:** Sonja Kuipers, Stynke Castelein, Linda Kronenberg, Job van ’t Veer, Nynke Boonstra

**Affiliations:** 1Research Group Healthcare & Innovation in Psychiatry, Department of Healthcare, NHL Stenden University of Applied Sciences, Rengerslaan 8-10, 8900 CG Leeuwarden, The Netherlands; 2Department of Experimental Psychopathology and Clinical Psychology, Faculty of Behavioural and Social Sciences, University of Groningen, Grote Kruisstraat 2/1, 9712 TS Groningen, The Netherlands; 3Lentis Research, Lentis Psychiatric Institute, Hereweg 80, 9725 AG Groningen, The Netherlands; 4Dimence Mental Health Care, Burgemeester Roelenweg 9, 8021 EV Zwolle, The Netherlands; 5Research Group Digital Innovation in Care and Welfare, Department of Healthcare, NHL Stenden University of Applied Sciences, Rengerslaan 8-10, 8900 CG Leeuwarden, The Netherlands; 6KieN Early Intervention Service, Oosterkade 72, 8911 KJ Leeuwarden, The Netherlands; 7UMC Utrecht Brain Center, University Medical Center Utrecht, Heidelberglaan 100, 3584 CX Utrecht, The Netherlands

**Keywords:** oral health, nursing, mental health, schizophrenia, psychotic disorder, human-centered design, personas

## Abstract

In mental health, oral health is often given little attention. Mental health nurses (MHNs) are professionally the appropriate target group to support maintaining and increasing oral health. We aimed to develop and validate personas that reflect the attitudes and needs of MHNs regarding oral health in patients with a psychotic disorder. We used a human-centered design with contextual interviews (n = 10) to address the key issues of the problems and needs of MHNs working with patients with a psychotic disorder. We analyzed the data thematically and reflected on insights into unique personas, which were then validated by conducting semi-structured interviews (n = 19) and member checking. Four personas were found based on attitudes and perspectives, barriers, needs, suggestions for interventions, and site conditions regarding practicing oral care in this patient group. Our findings were as follows: the attitudes and perspectives differed from not feeling any responsibility to a holistic obligation, including oral health; suggestions for interventions for MHNs ranged from interventions focusing on improving skills and knowledge to using practical tools; most MHNs recognized themselves within a persona that had a holistic obligation that included oral health; in addition, the MHNs indicated that they considered the issue of oral health in this patient group important, but, in practice, took little responsibility for that role. These findings suggest that a toolkit with interventions for MHNs that are tailored to the personas that emerged from our research should be developed by MHNs in co-creation with designers. The differences between the perceived role and MHNs’ practice in oral health highlighted the need for role clarification and professional leadership of MHNs regarding oral health, which should be considered when developing interventions.

## 1. Introduction

Since the World Health Organization emphasized that oral health is integral and essential to general health and wellbeing [1,2], oral health has improved in the general population; however, vulnerable patients are an exception to this [3]. For instance, poor oral health in patients with a psychotic disorder may lead to poor self-image, low self-esteem, decreased self-confidence, social phobia, loneliness, depression, and suicidal intent; these people are ashamed and are afraid to go outside, and therefore, participate less in society [4,5]. Epidemiological studies showed that the lifespan of patients with a psychotic disorder is shorter than that of the general population without mental illness. The gap in mortality was estimated to be a 15–25 year shortened life expectancy in patients diagnosed with a (severe) mental illness, including in countries where the quality of healthcare is acknowledged to be good [6,7]. An unhealthy lifestyle is an important cause of the gap in mortality. Several studies among patients with (severe) mental illness showed that oral health and oral-health-related quality of life are substandard within the unhealthy lifestyle domain [8,9,10,11,12,13].

Inadequate oral health self-management, a lower tooth-brushing frequency, a lack of motivation for proper oral hygiene, and poor psychosocial functioning are known as other barriers to adequate oral health in this patient group [14,15,16,17]. From a holistic perspective [18], supporting the general health of patients with a psychotic disorder, including oral health, is one of the tasks of a mental health nurse (MHN), which is one of the main health professions at the forefront of everyday care services [13,19]. MHNs indicate that they hesitate to take action and would like to be more attentive in this area [13]. MHNs state that they lack the relevant expertise, and there is a lack of practical interventions for MHNs to use when supporting patients regarding oral health. Happell et al. [19] discussed the importance of this topic being included on the agenda in mental health organizations, which are often only focused on psychiatric and psychological problems. One of the barriers reported was that when patients express concerns regarding their physical health, these concerns tend to be given little importance by the healthcare professional [19]. Furthermore, MHNs in general have to provide more input on needs and barriers in mental health services.

Dutch and British guidelines regarding people with mental health problems [14,20] do not meet the needs of the MHNs due to the lack of intervention options. Therefore, MHNs feel that it is inconvenient to uphold what these guidelines prescribe. A recent scoping review examined the educational, behavioral, and physical interventions to improve oral health among patients diagnosed with a mental health disorder [11]. An important conclusion of this review was that, despite the importance of good oral hygiene, few interventions have been developed for MHNs. Although the interventions that do exist were shown to be effective in the short term (<1 year), these interventions were developed without the involvement of the end users of these interventions (i.e., mental health professionals). It is unclear what has been missed while developing these interventions. An important reason the development of new interventions is often unsuccessful lies in the fact that developers of (medical) interventions do not thoroughly understand the perspective of the end users. Interventions are often developed on a theoretical basis without actively involving the target group [21]. Many organizations fail to consider the MHNs as end users of interventions, and therefore, the starting point of the design process [22]. Thus, interventions to support patients with a psychotic disorder should address the needs of MHNs as end users. Therefore, MHNs, as well as experts by experience, should be involved in the development of interventions. The involvement and engagement of MHNs as end users will increase the chances of developing effective interventions [23].

The current research project was undertaken as a first step in a longer design-oriented project that aims to design mental health nursing intervention(s) for oral health care in patients with a psychotic disorder. Specifically, we sought to increase clarity regarding the attitudes, barriers and needs, and suggestions of MHNs to provide support for maintaining and increasing oral health in patients with a psychotic disorder. The outcomes of this study might lead to validated personas with useful insights into MHNs regarding maintaining and increasing oral health. These personas can serve as an empathic handover in the development of oral health nursing interventions [24]. 

## 2. Methods

### 2.1. Study Design

For this study, a qualitative, descriptive, interpretative design was conducted to gain insight into the attitudes, barriers and needs of MHNs regarding oral health in patients with a psychotic disorder. To determine the contributions to practice, a human-centered design (HCD) approach was adopted. An HCD approach is a co-creative, iterative, and creative approach with non-linear steps to problem-solving to provide tailor-made solutions for severe problems [25]. This participatory and iterative HCD approach differed from other methods by empathizing with MHNs from the start of the project and working in co-creation with MHNs to develop a deeper understanding of their needs and, therefore, design more suitable interventions and give them a decisive voice in solution directions [26,27]. One widely applied HCD approach for innovative design projects is the Double Diamond (DD) framework, which was developed by the Design Council [28]. This DD framework guides problem-solving thinking in the HCD design process. The current study focused on the first, exploratory, steps in this process (the “discover” and “define” phases of the DD framework) aimed at one major task: the intervention must address the key issues of the problem, and thus, the needs of the end users [28]. The iterative stages in the research process are shown in Figure 1. The participation of MHNs throughout the design process has several purposes: (a) at the start, the participation of MHNs helped us to explore the context of mental health professionals, including their worldviews and needs; (b) the MHNs could participate in constructing and/or utilizing prototypes; (c) the MHNs could give feedback in user tests; and (d) the MHNs could give insight into the system in which an intervention should be implemented and the needs of MHNs in caring for patients with a psychotic disorder [29,30,31]. 

### 2.2. Population

In one of the first steps (step 1a), we conducted contextual interviews (Figure 1). To achieve maximum variation in our sample, a representative sample of participants was selected by the research team [32] based on their knowledge, years of experience with patients with a psychotic disorder in all stages [33], and affinity with oral health. In this step, various levels of MHNs were included: bachelor’s-level mental health nurses, bachelor’s students of mental health nursing, master’s-level advanced nurse practitioners, students of the master’s degree of advanced nurse practitioners, experts by experience, and oral health hygienists. Participants were approached face-to-face. 

In the second step (step 2a), we conducted semi-structured interviews with MHNs (Figure 1). A convenience sample was obtained using snowball recruiting based on experience (in working years) with patients with psychotic disorders, availability, and willingness to participate. Participants were approached by email or face-to-face. Participants were recruited from the KieN Early Intervention Service in Leeuwarden, students of the Bachelor of Nursing program in their last year (University of Applied Science NHL Stenden, Leeuwarden, The Netherlands), and students of the Master Advanced Nursing Practitioner program (University of Applied Science GGZ-VS Utrecht). 

**Figure 1 ijerph-20-03475-f001:**
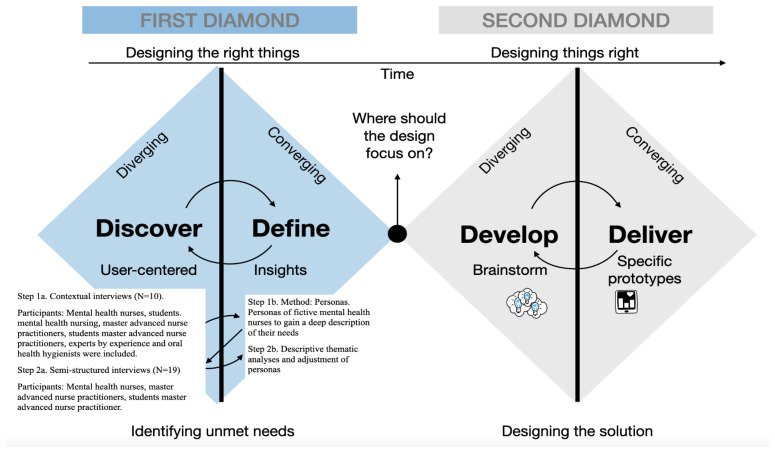
The iterative HCD stages and activities used in the first diamond of the Double Diamond model from the Design Council [28], adapted with permission from van ’t Veer, Wouters, Veeger, van der Lugt, 2021 [34]. Copyright 2020 by publisher Coutinho, Bussum, the Netherlands. Note: HCD (human centered design) stages and activities in the two diamonds of the Double Diamond model [28]. Steps 1a–2b describe each stage of our project (discover and define). As HCD is an iterative process, the arrows describe how the results of each step impact the next step, leading to a detailed description of four different personas regarding oral health and adjustment of the personas in the last step (step 2b).

### 2.3. Data Collection

#### 2.3.1. Contextual Interviews

For the contextual interviews (step 1a), we aimed to deeply understand (1) the attitudes and daily practices of MHNs regarding oral health, (2) the obstacles and barriers MHNs experienced related to oral health in daily practice, (3) what suggestions MHNs had in terms of oral health nursing interventions, and (4) what conditions are required, according to MHNs, to support their patients with a psychotic disorder by interviewing them in the context of their work [35]. The data were collected by a trained research nurse (S.K.) between January 2021 and June 2021. All interviews were audio-recorded and transcribed verbatim. 

#### 2.3.2. Semi-Structured Interviews

The purpose of conducting semi-structured interviews in step 2a was to gather information from participants who had experiences, attitudes, perceptions, and beliefs related to the topic of interest. These semi-structured interviews were used to (ecologically) validate the findings in the personas (step 2b) (see analysis, Section 2.4.2). In this case, ecological validity referred to the realism with which the persona matched the MHNs or their team members’ real work context and how accurately the personas reflected the relevant characteristics and their context in the world or environment [36]. Personas were characterized as valid when nurses recognized themselves or team members in a persona. The semi-structured interviews included a shortlist of “guiding” questions [37] (e.g., with which persona (or elements of the persona) do you identify most? Looking at the team of nursing colleagues you work with, which persona or elements are most recognizable? With which suggestions for interventions do you identify? Are these interventions relevant and useful in your work with patients with a psychotic disorder and why?) (Appendix C).

The personas (created in step 1b) were presented to MHNs and served as a compass during the semi-structured interviews. Participants were asked to read the persona descriptions, identify personal connections with the personas and characteristics, describe additional barriers and needs that influenced their opinion, and generate ideas for interventions and site conditions that would be useful to other mental health professionals that share their characteristics, barriers, and needs. 

The semi-structured interviews were conducted between January and April 2022. The interviews were audio-recorded and transcribed verbatim by the first author (S.K.). The iterative process of sampling, data collection, and analysis was continued until data saturation was reached.

### 2.4. Analysis

A thematic analysis [38] was used as a guiding procedure for the analysis of the data from the contextual interviews (step 1b) and the semi-structured interviews (step 2b). We analyzed the data using both a deductive and inductive thematic analysis approach [38]. In the discovery phase, structures and patterns became visible through fragmentation. The fragments were given open codes. The reduction phase was devoted to identifying coherent themes based on the codes. The themes were then revised and refined. In the last phase, we reflected on the themes and the analysis process [38,39]. All data were analyzed in Atlas TI Version 9 (Atlasti.com). To improve the credibility of the results, we used member checking, where following a participatory and qualitative approach, the verbatim transcripts were returned to the participants to validate the data from the contextual interviews. Quotes were added to the themes to provide a rich description.

The key themes of attitudes and perspectives, barriers, needs, interventions, and site conditions were defined a priori for the analysis in step 1b, as they related to our research questions. Data coding facilitated the identification of patterns within the data and group statements with thematic similarities [39]. The themes and codes that emerged from the contextual interviews were used for persona creation. 

In step 2b, the semi-structured interviews were analyzed using thematic analysis [38] as a guiding procedure to ecologically validate the findings in the personas. Data were analyzed inductively in terms of three themes: (1) identification with personas; (2) evaluation of barriers, needs, interventions, and site conditions; and (3) current oral care and the future of oral care in mental health care to validate the personas and to find additional information. 

#### 2.4.1. Persona Construction

In step 1b, the user insights provided through the contextual interviews were converted into four authentic and lifelike personas of MHNs working with patients with a psychotic disorder. According to Grudin and Pruitt, the creation and use of personas have extraordinary potential [27,40]. Personas are seen as a powerful tool for true participation in design, and they also force designers to consider social and political aspects of design that otherwise often go unexamined. In contrast to Grudin and Pruitt [27,40], Chapman et al. [41] has more skepticism about the persona method and discusses several methodological and practical limitations. For example, they point out the problem that it is hard to verify to what extent a persona can represents a larger (parts of) population [41]. In this research, personas were a part of the data-synthesis activity since it serves designers better have a specific person to empathize with when designing new interventions [42]. Pictures and background information were added to help designers to create an elaborate and relatable image of this person. Therefore, personas have the function of an “empathic hand-over” [24] by giving a living insight into the MHNs through their reading of the personas. 

To translate insights into useful design implications, personas are a useful method to present them. From the themes, patterns, and associated descriptive phrases in the transcripts, the researcher S.K. developed personas of typical MHNs related to oral care in collaboration with the research team (S.C., J.V., L.K., N.B.). The data within the contextual interviews were organized within five larger categories to highlight the MHNs’ (1) unique attitudes and perspectives regarding oral care, (2) barriers regarding oral health, (3) needs regarding oral health, (4) suggestions for interventions, and (5) site conditions needed to provide oral health to patients with a psychotic disorder. Based on the data, four types of nurses could be described as personas. Each persona was a composite portrait that incorporated representative data from participants. Relatable names and photos, illustrative quotes, and a description of their professional context based on the codes and patterns identified were also given to the personas [27,40]. 

#### 2.4.2. Persona Validation

In step 2b, the five-step Synthesized Member Checking (SMC) [43] was used for ecological validation and further development of the personas. SMC enabled the participants to add comments, which were then searched for confirmation or disconfirmation of engagement with the personas, enhancing their credibility [43]. Discussion about participants’ feedback and changes were incorporated into the personas. 

To improve dependability and confirmability, an audit trail was conducted whereby the raw data and memos generated during the study were saved in a log file. Data analysis of nine semi-structured interviews was performed independently by two researchers (S.K. and S.S.). Differences between both researchers while interpreting and coding the data were discussed until a consensus was reached.

### 2.5. Ethical Considerations

Standard rules for good clinical practice and ethical principles that have their origin in the Declaration of Helsinki were followed by informing all participants about the study and their rights, and all subjects gave oral informed consent [44]. For this study, according to the Dutch Medical Research with Human Subjects Law, we did not require ethical approval since professionals were included in the study. The recordings of the interviews were retained according to the international safety regulations for the storage of data at the NHL Stenden University of Applied Sciences and were accessible to three researchers of this project (S.K., N.B., J.V.).

## 3. Results

### 3.1. Contextual Interviews

We interviewed ten experts: four MHNs, two master’s advanced nursing practitioners, one student master’s advanced nursing practitioner, one student bachelor’s mental health nurse, one expert by experience, and one oral health hygienist. The group consisted of eight women and two men; the ages ranged between 23 and 52 years (mean age: 38 years). The number of years of work experience varied between 0 and 34 (mean: 9.3 years). Participant characteristics (profession, gender, age, educational level, and years of working experience) are given in Appendix A. It is important to notice that themes were linked to each other (e.g., needs appeared from barriers, interventions were associated with needs, and site conditions also referred to needs and barriers). 

#### 3.1.1. The Attitudes, Perspectives, and Daily Practices of MHNs Regarding Oral Health 

The attitudes and perspectives of oral health in mental healthcare and what a mental health professional is supposed to do about oral care varied enormously among the teams of participants. Of the participants, four stated that MHNs should include oral health in their repertoire of actions. Out of the ten participants, two participants discussed oral health care in their team and stated that their colleagues did not consider oral health care to be part of their job and that the role of the family should be more prominent. The expert by experience supported this finding: 

*Because in my own experience with my bad teeth in health care, it was sometimes signalled by MHNs, but then nothing was done about it. And that is because, well, people (MHNs), do not think that is their job. And that is a wrong way of thinking*.  (Participant 9)

The other participants did not feel responsible for maintaining and increasing oral health as part of their daily work. 

#### 3.1.2. Barriers to Providing Oral Health Care

The barriers participants experienced were quite diverse. The MHNs stated that when patients were admitted with a psychotic disorder, they acknowledged that there was a lack of materials (no toothbrush or toothpaste) or the MHNs did not know what was available. Sometimes, this was noticed after six days of care and patients were not asked. When patients are admitted with a psychotic disorder, care is focused on clinical recovery (e.g., reduction or cessation of symptoms). Oral health care is not considered part of recovery, and there is no focus on physical issues (e.g., oral health). 

In mental healthcare, it is felt that every patient should perform their activities of daily living, and little attention is paid to personal care at all. Participants often did not know whether there was an assurance for oral health care. Additionally, participants reported not having the time to provide oral health care. 

*We have about 30 min with a patient. In that time, we also must check on medication, and symptoms. After 30 min, we should go to the next patient. There is no time for MHNs to do something with oral health*. (Participant 2)

Most participants stated that the attitude of “a lack of attention for oral health” is not good. Participants stated that there was a lack of knowledge on what to do and why. Participants also indicated a lack of knowledge regarding what kind of information patients should have. It is well-known that salivation is a side-effect of anti-psychotic medication; the participants told patients that they must drink a lot of water. However, this also contributes to caries and dental erosion, which was not known, as reported by two participants, and therefore, not told to patients. 

One participant said that she had a patient with an oral health burden. Sometimes, due to the negative symptoms of psychosis, the patients are not motivated, are lethargic, and there is a lack of initiative. The participants saw that patients often felt insecure about their oral health (e.g., by talking with a hand in front of the mouth), but the participants did not know how to incorporate this into daily care. All participants indicated that there was too little knowledge and awareness about oral care within the teams and that this was the main reason they did not know what to do. 

One participant stated that she really did not know how to motivate and stimulate patients. Mental health staff downgraded these problems because oral health problems are physical and their patients are admitted for psychological or psychiatric problems. 

The oral hygienist named the high costs for patients when there is no additional oral health insurance as the most important barrier. When patients contact an oral hygienist, there is often a lot of burden and treatment is only reactive; regular appointments and prevention were noted as important. 

*The cost of treatment of severe symptoms is too high for patients and a barrier to make an appointment. We see patients too late*. (Participant 6)

*In practice, at the time of intake, we know what medication patients are taking and we can respond accordingly. But if something changes in the meantime and we are not informed, it becomes difficult*.(Participant 10)

#### 3.1.3. Needs of MHNs for Providing Oral Care 

Within the theme of needs, we found the subthemes of knowledge, which contained statements about the knowledge that the participants needed, and skills, which contained statements about the practical competencies the participants needed. 

Overall, some participants (MHNs and the expert by experience) seemed to be unconsciously incompetent. Participants acknowledged their lack of knowledge regarding oral health. Half of these participants finished the Bachelor of Nursing without any education on oral health care. Most frequently mentioned in terms of lack of knowledge were the importance of oral health, oral health in mental health, oral health diseases and symptoms, and medication and oral health. 

*I know that oral care is important, but I do not know why. Well, brushing twice a day, I think that when I have more background information and underpinning information, I will also do more with it in practice*.(Participant 3)

All participants stated that the need for skills stemmed from a knowledge deficit. Some participants indicated that observing oral health, being able to ask questions, explaining properly, and assuming a coaching role are important skills, but they are not always applied. In terms of skills, all participants agreed that effective communication skills are needed. Half of the participants confirmed that these skills were needed to talk to patients about oral health. Motivational interviewing is an important part of this, but according to one participant, MHNs have mastered the skill of motivational interviewing but do not know how to apply this to oral health. Some of the participants indicated that they felt the need to know how best to start a conversation about oral health. Other participants stated that they can start a conversation but are unsure of what actions to perform afterward. 

*Good communication skills are essential, because the shame and the fear of stigmatization play a crucial role, so you must be able to empathize with patients*.(Participant 9)

#### 3.1.4. Interventions for MHNs for Providing Oral Health Care

During the interview, the participants were asked what they needed in terms of interventions regarding oral health. Less than half of them said they needed training as an intervention. Training can occur in the form of a continuing education course, refresher course, or peer review to gain more knowledge and create more awareness about oral health. Participants indicated that a one-day training could contribute to more knowledge and create awareness. 

Some participants would like to give patients an information sheet or flyer (e.g., after the intake or during a care plan meeting), or hang leaflets above the sink in rooms. Three participants (MHNs and the oral hygienist) introduced the importance of including oral health in an intake/anamnesis/somatic screening (e.g., an oral health screening) and care planning (and care coordination meetings) for patients. The integration of an oral health screening during patient admission would allow MHNs to include oral health care in treatment plans.

*I see that also that fellow MHNs do not concern themselves with this, and when patients are admitted to an outpatient-team (e.g., early intervention psychosis team or assertive community treatment teams), no attention is paid oral health, and a somatic screening is done and patients were physically examined, except in the mouth*.(Participant 3)

One participant mentioned the importance of lifestyle within mental health. Lifestyle was seen as highly prioritized in organizations. The MHNs gave training and education to patients regarding lifestyle, but these were related to tobacco, alcohol, drugs, and physical activity; the education was less focused on hygiene and oral health as part of lifestyle training. 

Participants proposed that an app could be used to provide oral health advice. Additionally, the oral hygienist advised the MHNs to develop a digital decision tool for oral health that contained all the required information in it. 

*A digital decision app with knowledge on oral health (e.g., importance, risk factors) and an advice nurses can give to their patients (e.g., based on oral health screening) could be appropriate, like a decision tree*.(Participant 10)

#### 3.1.5. Site Conditions to Provide Oral Health Care 

Some participants indicated that not all patients have the right tools to take care of their oral health properly. It would be nice if tools were available and within easy reach. Tools, such as toothpaste, a toothbrush, and mouthwash, would be nice to have within reach for some participants. Two participants (a mental health nurse and an expert by experience) suggested that patients received a welcome pack when they are admitted, which could include a small tube of toothpaste and a toothbrush.

Almost all participants stated that interventions should be facilitated by the organization. In this way, an intervention is better secured and all patients have the same starting point. Organizations must support MHNs and make it clear that it is a part of MHNs’ day-to-day work. Participants felt they needed to have more time to observe patients and give good oral health information or education within an organization.

### 3.2. Personas

From the contextual interviews (n= 10), four unique personas of MHNs were identified: three women and one man (Figure 2, Figure 3, Figure 4 and Figure 5). These persona descriptions contained interpreted data from the contextual interviews. The results were related to the nurses’ (1) attitudes and perspectives on oral health, (2) barriers, (3) needs, (4) suggestions for interventions, and (5) site conditions.

### 3.3. Ecological Validation of the Personas

Subsequently, semi-structured interviews (n = 19) were conducted with MHNs (n = 6), master’s advanced nursing practitioners (n = 5), student master’s advanced nursing practitioners (n = 7), and a student mental health nurse (n = 1). We interviewed fifteen women and four men; the ages ranged between 22 and 54 years (mean age: 37). The number of years of working experience varied between 0 and 34 years. All participants worked in a clinical or outpatient center with patients with a psychotic disorder in various stages. The participant characteristics (profession, gender, age, educational level, current team working, current patient category, and years of working experience) are given in Appendix B.

#### 3.3.1. Identification with Personas

We found a resemblance between the participants’ attitudes and perspectives regarding oral health in patients with a psychotic disorder, barriers, needs, suggestions for interventions, and site conditions, as presented above, and their origins in the personas. When participants looked at their role in the context of their work, the MHNs identified with Anna (n = 9), Julia (n = 6), or combinations of elements in Anna and Julia (n = 8), and with Julia and Paul (n = 1), indicating that most nurses supported the holistic perspective of oral care and the integration of oral care in daily care (Appendix D, Appendix E and Appendix F). All the participants felt that addressing the oral health of patients with a psychotic disorder was important. Participants perceived oral health activities to be within the role of MHNs. However, there was a lack of consensus within participant teams; colleagues did not always perceive oral health to be the role of MHNs. 

An often-shared perspective on oral health was Anna’s view on oral care: “It is important to look at oral care from a holistic perspective”. According to participants, this perspective suits recovery-oriented mental health practice and person-centered services. Julia’s perspectives were in line with Anna’s perspectives, i.e., the integration of oral health care in general care is part of a holistic perspective. 

*But we, as MHNs, do not practice oral health*.(Participant 5)

The participants unanimously stated that Monica’s perspectives were not appropriate for a mental health professional. Almost half of the participants did slightly agree with this persona in that they expected that a shift in attitude could be the result of proper training in oral health. In addition, a beginning nurse needs to have a good example when starting a career in mental health care. If this is the attitude of the whole team, this might explain the attitude of a beginning nurse.

*Monica’s perspective is based on the medical model, which focuses on (the absence of) psychiatric complaints. That is not how it works. But also, her limited experience might be an underlaying reason for her current attitude. That means she requires a proper role model*.(Participant 7)

When the participants looked at the team of nursing colleagues they worked with, most participants recognized all personas. One of the participants stated that almost half of their team was made up of people like Monica, but the participants did not recognize themselves as a Monica. 

#### 3.3.2. Evaluation of Barriers, Needs, Interventions and Site Conditions

Overall, the participants recognized almost all barriers, needs, interventions, and site conditions experienced in the different personas, in themselves, or in the teams they worked in (Appendix G). This indicated that there was no one-size-fits-all method for MHNs to maintain and increase oral health in patients after a psychosis admission.

Barriers are often linked to site conditions. Following an episode of psychosis, patients are often not motivated to maintain or increase oral care. It is difficult for MHNs to motivate patients in terms of maintaining their oral health, partly because patients expect to be treated for mental health problems. To start talking about oral care is perceived by patients as strange because they do not consider it as part of their treatment. This makes it challenging to initiate a conversation about oral care with patients. 

The participants stated that since oral care is not asked about during intake, the MHNs acted only reactively when problems arose and were not proactive enough to prevent problems.

*This is so recognizable. Not all patients have a toothbrush and toothpaste. I had a patients who had three tubes of toothpaste and no toothbrush, our team found this out three months after the patient’s admission. Nobody does anything about this*. (Participant 2)

*Today, I had a 28-year-old boy who came to me because he had gum problems. He had not brushed his teeth for a long time. He did not have any materials with him, but we have them. We must show more initiative in an earlier stage because he had already been on admission with us for 9 days*. (Participant 18)

Participants stated that short videos should be added to the interventions in the persona of Julia. 

*Short, 2 min videos would be useful. If I must do something at home (e.g., adjust a derailleur for a bicycle), I quickly look at YouTube, then you can see what needs to be done. And it is relevant to quickly see an example*. (Participant 1)

#### 3.3.3. Current Oral Care and the Future of Oral Care in Mental Health Care

All participants indicated that they currently do nothing to improve or maintain oral health in patients with a psychotic disorder. When oral health is discussed with patients, participants perform these activities on an individual basis, but follow-up is lacking. When looking at the future of oral care, the participants mentioned educating students in nursing and mental health professionals in knowledge and skills to raise awareness, integrating screening into the anamnesis so that the dialogue can also be about oral health, ensuring there is a follow-up in treatment plans, and that organizations should facilitate oral care by ensuring that materials are available. Maintaining and increasing oral health should be a topic of conversation within MHN teams. 

## 4. Discussion

The current study aimed to gain insight into the attitudes and needs of MHNs regarding support for maintaining and increasing oral health in patients with a psychotic disorder to guide the development of interventions for this population. Our main findings were as follows: (1) there was diversity in attitudes and perspectives on oral health from MHNs; (2) there were differences in barriers, needs, and suggestions for interventions from MHNs; (3) in contrast with the fact that almost no attention was given to oral health in daily practice, MHNs recognized themselves in a persona that encouraged a holistic perspective that included oral health; and (4) in line with the MHNs’ holistic professional profiles, most MHNs indicated that they considered the issue of oral health in patients with a psychotic disorder important, but in practice, they took limited responsibility for that role.

To elaborate on the first finding, we found that the attitudes and perspectives of the MHNs regarding oral health in patients with a psychotic disorder varied enormously. Although some MHNs suggested that oral health in mental health is important, the findings suggested a difference between these expressed beliefs and the actual behavioral follow-up in daily practice. There is still a struggle for MHNs to take responsibility for oral health as part of general health in patients with a psychotic disorder. The literature shows that MHNs’ attitudes toward addressing physical issues are positive in general, but MHNs need more education to effectively include physical health promotion in their activities [45]. A recent study [46] showed that existing psychiatric care needs were well discussed by psychiatrists, psychologists, and MHNs, but physical care needs and social-wellbeing-related care needs remained untreated in psychosis care. Happell et al. [18] showed the need for education and training for MHNs to improve physical health care, but oral health as part of physical health was not included. The results of this study showed that some MHNs felt responsible to act on oral health issues, and several MHNs in certain situations felt responsible but were still uncomfortable acting on oral health. Oral health issues are important, as related risk factors are known to contribute to a lower quality of life and reduced life expectancy [8,9,10,11,12,13]. Since the MHN profession is underpinned by the concept of a holistic vision of care [18,47,48], it is important that the integration of physical issues, such as oral health, is addressed as a matter of priority. 

To elaborate on the second finding, there were differences in the barriers, needs, and suggestions for interventions among MHNs. The most important barriers acting against the MHNs to play an active role in oral care for their patients were a lack of information (e.g., what is proper oral health care? What is the benefit of proper oral health care for patients? What are the risk factors? What do MHNs have to look out for?), a lack of practical skills (e.g., how to carry out oral health care in patients), and a lack of practical facilities (e.g., an oral health screening form). However, the most important barrier was reported to be the patients, since they are often not capable of improving their oral care. This is in line with the literature [19,49]

The MHNs suggested the development of a wide range of different interventions for the MHNs to practice oral health in patients with a psychotic disorder, including interventions focusing on improving knowledge, improving skills, practical tools, and improving motivation in patients. 

For MHNs with personas like Monica, Julia, and Paul (who reported a need for educational interventions to increase knowledge and awareness, as well as practical skills regarding oral health in their patient group), oral health training should be developed. A pre–posttest and a scoping review [11] showed that educational interventions were effective at improving oral health knowledge in MHNs of patients with SMI. The results of these studies are promising and can serve as a rationale for the development of educational interventions in the future. Furthermore, an appropriate method needs to be found to substantiate, disclose, and valorize good interventions.

MHNs like Anna, Julia, and Monica need more information on oral health (e.g., the influence of different anti-psychotics on oral health for MHNs like Anna and Julia, and information about the advantages and disadvantages of oral hygiene for Monica); it was suggested that leaflets and posters could be tools to provide oral care information to MHNs and patients with a psychotic disorder. A systematic review of oral health promotion in a general community setting showed that traditional health promotion tools, such as leaflets and posters, are useful, especially for adults [19]. MHNs like Julia need additional practical information (e.g., an example of how to brush). This systematic review was focused on the effectiveness of interventions; the content (e.g., topics or development in co-creation) of oral health promotion was not known. Therefore, oral health promotion tools, such as leaflets and posters, should be developed for patients with a psychotic disorder in co-creation with MHNs. Leaflets and posters are more effective when combined with other media, such as videos [50], and thus, the suggested 2 min videos could be supplementary to such tools. Patients and MHNs can profit from these short videos by embedding these videos as micro-learning intervention in a mobile application [51].

The presence of MHNs like Anna suggested the need for oral health psychoeducation for patients with a psychotic disorder because this would allow MHNs to educate their patients on the importance of oral health and to engage patients in how to integrate oral health care into daily routines. A Cochrane systematic review showed that psychoeducation is effective for knowledge provision in patients with SMI [52]. In mental health, psychoeducation is widely deployed. To date, however, there has been no psychoeducation available on oral health in patients with a psychotic disorder. 

MHNs like Monica, Julia, and Paul probably need an oral health screening form to screen a patient’s oral health status because they need a tool to integrate oral health outcomes into daily care to support patients with a psychotic disorder. A recent systematic review of oral health assessments for non-dental healthcare professionals discussed 18 different screening forms [53]. This review found that the Oral Health Assessment Tool (OHAT) is the best validated and most complete tool for use by non-dental professionals (such as MHNs) assessing oral health [53,54,55]. The OHAT was validated for use among senior care dependents in community dwellings [53]. This means that the OHAT is potentially appropriate and it might be preferable to specifically validate an OHAT for patients with a psychotic disorder. The development of a digital oral health screening tool can fit in the needs of MHNs. A recent review shows that the use of mobile health (MHealth) is promising for patients with a psychotic disorders [56]. However, considering this, it is important to discuss the impact of mobile technologies on the professional relationship between MHNs and patients with a psychotic disorder. Here, the findings of Schneider-Kamp and Fersch [51] demonstrate that technological solutions such as mHealth can improve some care processes for mentally vulnerable groups, such as patients with a psychotic disorder, and show how MHNs can get and stay involved in patients’ lives and everyday practices. Thus, the use of MHealth can support the reinvestment of time savings into the improvement of the MHN-patient relation. Additionally, it is important to point out that, for the positive effects of detached co-involvement to emerge and to avoid repercussions, a fine balance must be struck between face-to-face and MHealth [51]. MHNs like Anna need more information on motivation and engaging patients in oral health because her patients have told her that they are not motivated to engage in oral care, and Anna did not how to act. MHNs with these needs should use interventions that combine behavioral sessions (brief motivational interviewing sessions) and education sessions. This combination of interventions was shown to be effective for oral health (Q.H. plaque index and oral health knowledge) in patients with severe mental illness [11,57]. Therefore, the development of interventions that combine behavioral and educational elements to support MHNs to motivate and encourage patients with a psychotic disorder should be prioritized. 

Moreover, interventions suggested by the MHNs were shown to be effective, but there are important differences in context and population. This means that insights from these studies, while respecting differences, should take them into account during the design process when developing oral health interventions in co-creation with this patient group, MHNs, and designers. 

Providing oral care to patients with a psychotic disorder to maintain and increase oral health is an important role for MHNs, as oral health in mental health is part of the holistic perspective of nursing [18]. However, the availability of tools alone does not automatically encourage a behavioral change in MHNs regarding maintaining and increasing oral health in patients with a psychotic disorder. Therefore, it is important to develop interventions in co-creation with MHNs to obtain input on the appropriate content, timing, and scope of these interventions so that they fit into their workflow as well as possible.

We also found that, in contrast with the fact that almost no attention is given to oral health in daily practice, the MHNs recognized themselves in a persona that encourages a holistic perspective that included oral health. We asked participants about which persona they identified the most with. The MHNs frequently reported that they recognized the persona of “Monica” (e.g., not concerned with oral health, focus on psychological or psychiatric issues, no awareness) in colleagues, demonstrating its validity and the significant prevalence of this type of colleague. We did not find any MHNs who recognized themselves as a “Monica” and were able to formulate the needs of “Monica” (e.g., raising awareness and gaining more knowledge). It remains unclear whether no Monica-type professionals exist, or whether participants felt too uncomfortable to associate themselves with a person that represents a rather negative attitude. Due to the high variation in participants and the results of this study, there were no indications this affected the results. 

Lastly, in line with MHNs’ holistic professional profiles [18], most MHNs indicated they considered the issue of oral health in patients with a psychotic disorder important, but in practice took little responsibility for this role. These personas show nuanced differences that should be considered when developing interventions because they will encourage MHNs to reflect on what they are doing and to take professional leadership. The results of this study showed the need for role clarification [45] and professional leadership. Professional behavior is defined as personal leadership, such as acting proactively, role modeling, taking initiative, self-reflection, showing assertiveness or courage, and being focused on good cooperation [58]. Each MHN should be able to recognize and reflect from their professional leadership on what kind of information or skill needs they have to act better. This fits into the professional nursing profiles. The question of how MHNs can take a professional leadership role and take responsibility for their role as MHNs is highly relevant in this regard, but this question cannot be answered based on this study alone. Further research among MHNs regarding MHN professional leadership related to maintaining and increasing oral health in patients with a psychotic disorder is indicated. 

### Strength and Limitations of this Study

A strength of our study was the use of an HCD approach with a rigorous yet flexible research process, leaving ample room for adaptation; this provided an in-depth understanding of the attitudes and perspectives, and the barriers, needs, and suggestions, of MHNs. Therefore, the MHNs were engaged in the entire process and all stages of this project. Our personas are the first personas regarding oral health in MHNs that were validated and can be used to serve as an empathic handover in the following phases of the design process to develop oral health interventions. Additionally, these personas can serve as stand-ins for MHNs when team members have to make design decisions [59].

Within this design, we used two iterations (the first iteration was to develop the personas, while the second iteration was to validate the personas), and we used various populations and methods for triangulation, which helped to validate the findings by combining different methods, which is important to prevent fundamental biases that arise from the use of a single method (as in more traditional qualitative research). The wide variety of sources we used in this study, as well as the persona validation which provided a rigorous check on personas, increased the ecological validity. Additionally, Salminen et al. [60] discusses a lack of transparency as one of the challenges of creating personas, but we are confident that the rigorous described iterations show how the personas were generated and therefore decrease the lack of transparency. Our findings described the development and validation of four personas that can be used as an empathic handover and communicate with designers of management staff about the needs of MHNs regarding oral health. Based on data from a variety of sources, including the literature, contextual interviews, and semi-structured interviews, we identified and presented unique oral care perspectives, the main barriers, needs, suggestions for interventions, and site conditions, which were synthesized through four personas. These four personas were manually-created because there are no average or stereotypical MHNs [60]. 

From a strict scientific point of view, we recognize a possible issue with external validity [41]. However, persona creation may be considered more like a data-synthesis activity rather than a data-analysis activity. This means that, based on data, (design) researchers build personas to display relatable and authentic members of a target group, since it serves designers better to have a specific person to empathize with when designing new interventions [42]. Thus, although personas should be based on accurate data, ultimately, they serve a design process; they are not tools to meticulously represent a whole population in all its characteristics. 

Furthermore, our results showed that people recognized Monica in their work, but nobody recognized themselves as Monica. This can be explained as evidence that working with personas has added value: this may allow for the collection and interpretation of data in a way that reveals insights beyond social desirability. 

This study was carried out among a sample of MHNs distributed in various parts of the Netherlands (north, east, west, and south). The participants were representative in terms of sex, age, and educational level in nursing. While the number of interviews was modest, we were confident we reached data saturation, as no new information was retrieved in the last four interviews. The ecological validity of the personas and their elements was confirmed in this sample of participants after the second session of semi-structured interviews (n= 19), which meant that we were confident that widely recognized profiles of MHNs were established. One related issue in the creation of personas is that there are no empirically validated guidelines (e.g., as to how many personas should be created) [60]. For this research, we created four personas, data saturation and member checking showed that this was enough to give an answer on the research question in our population. Although we are confident that these profiles can be widely recognized, it is possible that we have missed profiles. However, these personas give an indication for the next step. 

The methods used in this study did not lead to the degree of empathizing that is required to achieve a greater understanding of the behavioral components of this problem, namely, professional leadership and taking responsibility. Further research regarding this behavioral component is required and should take professional leadership into account. For this, it would be preferred to use more generative methods (e.g., by deploying scenarios, having MHNs do things, or through observations) to get MHNs to think even more concretely about how they would act in each situation. Furthermore, the interventions suggested by MHNs were rather basic and the MHNs did not go beyond general interventions. The thoroughness of this research may have limited the speed of design-oriented research, but it was important to find out the content of these interventions. However, if we want to design effective, innovative interventions, it will be important to rely on MHNs and patients with a psychotic disorder in co-creation with a designer (educational, communicational) to broaden the scope in interventions.

## 5. Conclusions

When caring for patients with a psychotic disorder, physical care underpins psychosis care, and mental health professionals need to consider oral health care as an essential role in their daily tasks and provide necessary nursing support. MHNs need knowledge and awareness, screening forms, posters and leaflets, and access to short videos about oral care practices. MHNs need oral health psychoeducation and posters on oral health care to improve oral health in patients with a psychotic disorder. However, the development of interventions alone will not solve this problem. It is important to empower nurses and make sure that they collectively feel responsible for what is part of their job. 

The validated personas in this study can serve as an empathic handover while developing a toolkit with different interventions for MHNs and patients with a psychotic disorder. The next step in the HCD process should be, in co-creation with MHNs, patients with a psychotic disorder, and designers, the development of different oral health interventions in an oral health toolkit and an assessment of how this toolkit matches the attitudes and perspectives, needs, barriers, and suggestions expressed by the participating MHNs. However, within the concept of holism, the professional identity of MHNs and role identification are not independent of these factors.

Last, in this study, we described different strategies for MHNs to increase oral health in patients with a psychotic disorder. It is important to take these strategies into account while writing new guidelines (or editing existing guidelines) regarding oral health in patients with a psychotic disorder. 

## Figures and Tables

**Figure 2 ijerph-20-03475-f002:**
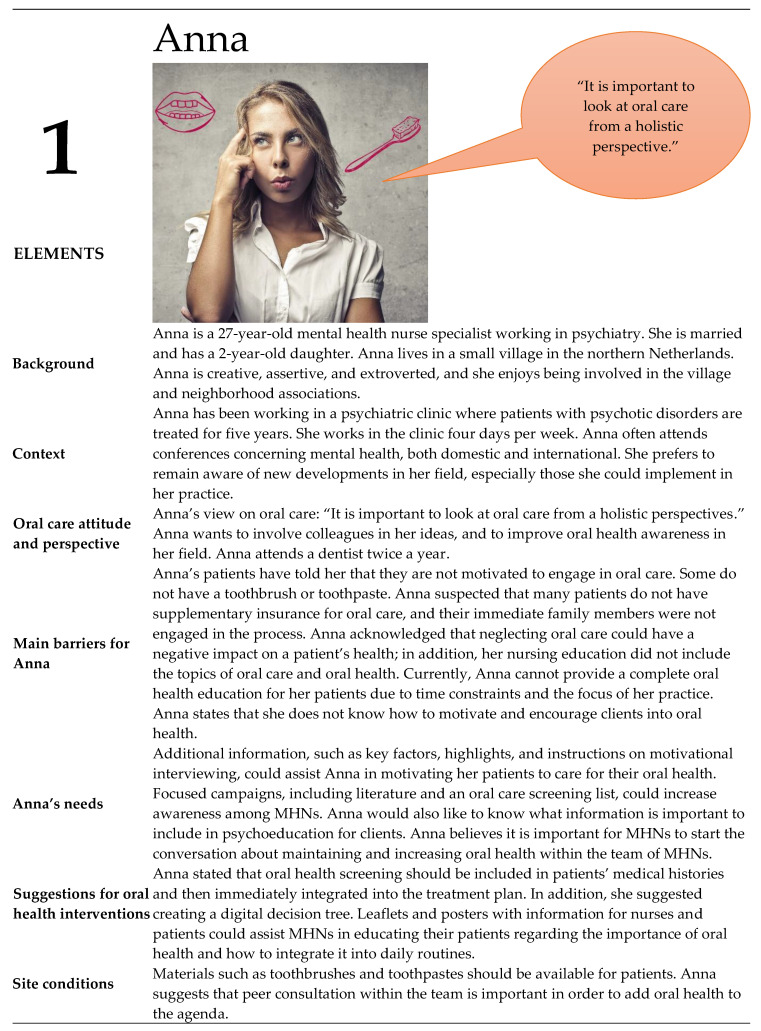
Persona Anna.

**Figure 3 ijerph-20-03475-f003:**
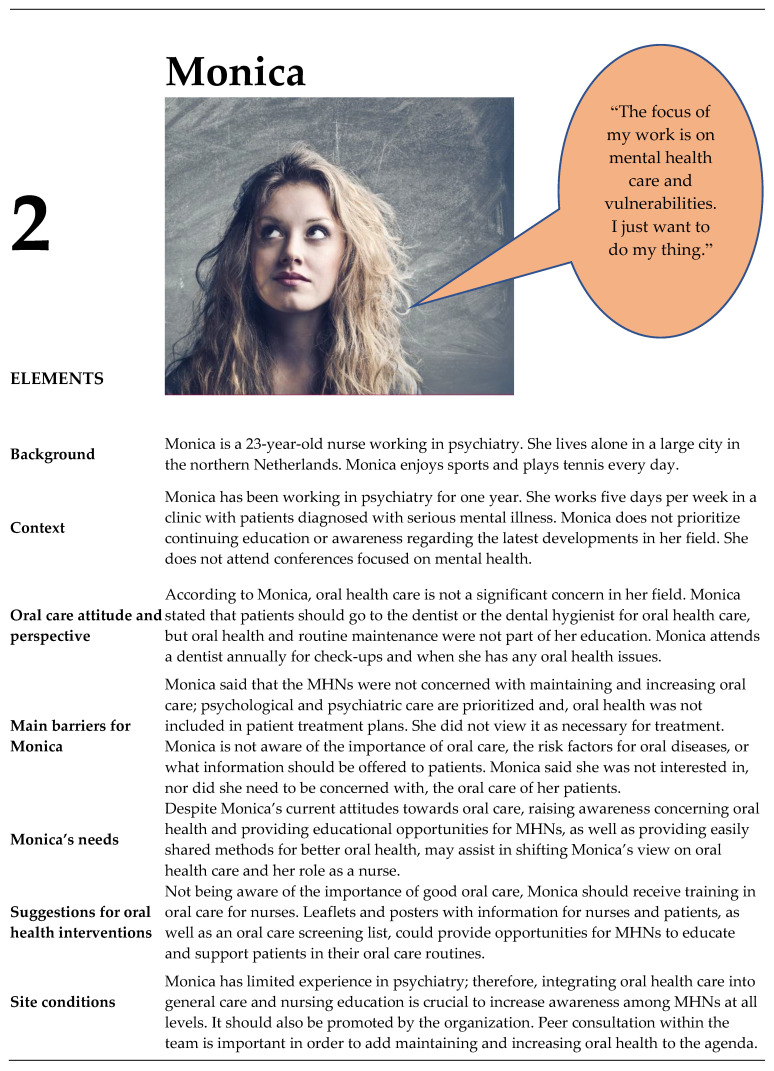
Persona Monica.

**Figure 4 ijerph-20-03475-f004:**
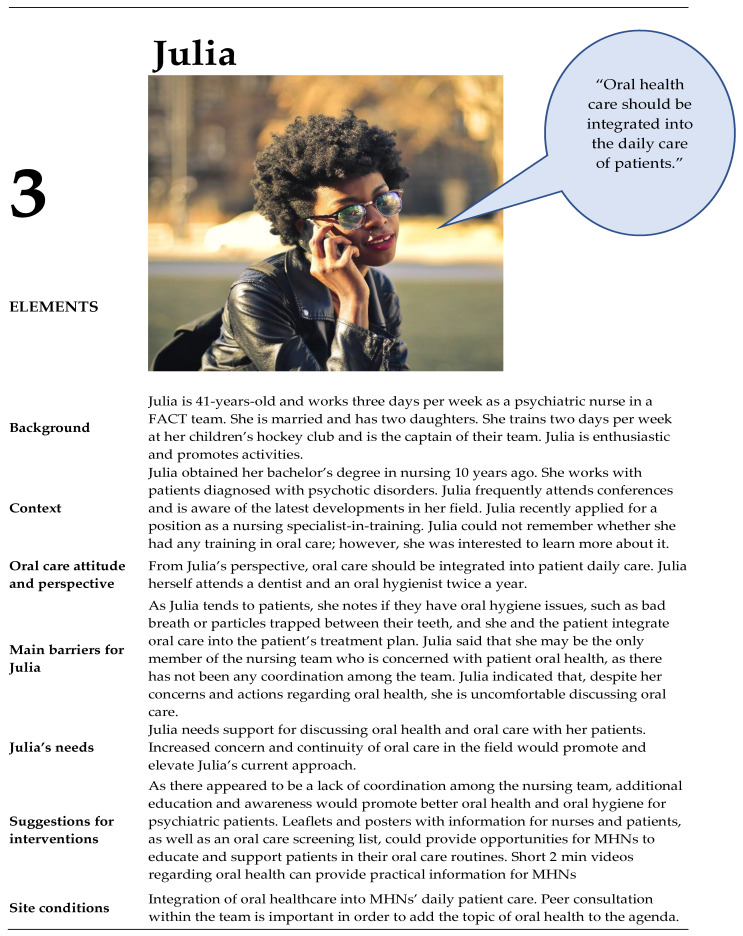
Persona Julia.

**Figure 5 ijerph-20-03475-f005:**
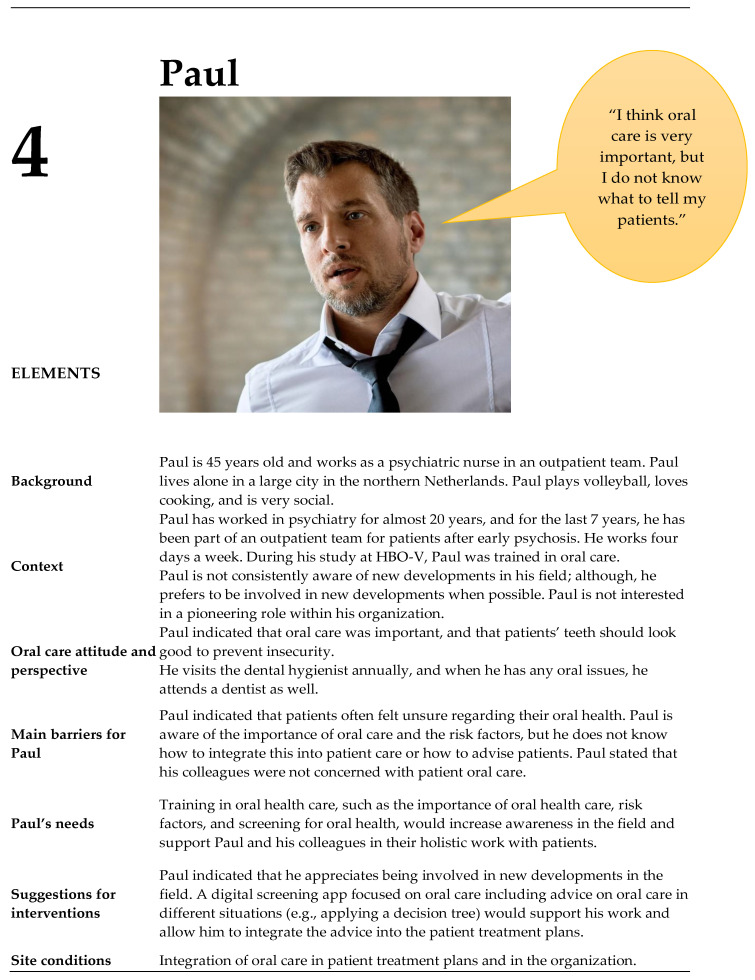
Persona Paul.

## Data Availability

Data is available from corresponding author upon request.

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
