# Peer review of "A Human-Centered Design Approach to Develop Oral Health Nursing Interventions in Patients with a Psychotic Disorder"

_ijerph, 2023, doi:10.3390/ijerph20043475_

Round 1

Reviewer 1 Report

Thank you for an interesting read. While the context of psychiatric disorders is not my main field of interest, I found the work highly relevant to the on-going debate on how to care for vulnerable patient groups. That said, the manuscript needs to be improved in two major aspects to be considered for publication.

First, the methodological approach of persona generation from qualitative data might or might not be appropriate. The manuscript currently only takes a positive view on personas, ignoring the widely spread critical voice that have surface in the wake of the popularity of personas in design. In the detailed comments below, I make some recommendations with which works to engage and where and how to improve the manuscript.

Second, the discussion currently reads more like a summary of findings with a few managerial implications. The manuscript needs to be improved by engaging with more of the extant work on the social dimensions of care for psychiatric patients, in general, and the use of digital tools, in particular. The detailed comments suggest some works to engage with in order to improve this aspect of the manuscript.

Detailed comments:

Line 24: Maybe "one or more psychotic disorders"?

Line 208: You only cite proponents of personas. There has been a lot of critique, which you should at least mention and position relative to:
Chapman, C. N., & Milham, R. P. (2006). The Personas’ New Clothes: Methodological and Practical Arguments against a Popular Method. Proceedings of the Human Factors and Ergonomics Society Annual Meeting, 50(5), 634–636. https://doi.org/10.1177/154193120605000503
Salminen, J., Jung, S.-G., & Jansen, B. J. (2021). Are data-driven personas considered harmful?: Diversifying user understandings with more than algorithms. Persona Studies, 7(1), 48–63. https://doi.org/10.3316/informit.352977339951659

Line 217: It might be more appropriate to use the term "categories" or "overarching categories" instead of "large clusters".

Line 226-229: You should keep to the reference style with the square-bracketed numbers.

Line 234: What do you mean by "discussed until a consensus was reached"? Please elaborate.

Lines 357-362: Here, you mention suggestions of developing a mobile app that support oral health for patients with psychiatric disorders. You do, however never come back to this in the discussion. I would suggest to amend the discussion on the co-creation of interventions in Lines 685-688 by engaging with current literature on mobile app-based solutions for supporting patients with psychiatric disorders:
Chivilgina, O., Wangmo, T., Elger, B. S., Heinrich, T., & Jotterand, F. (2020). mHealth for schizophrenia spectrum disorders management: A systematic review. International Journal of Social Psychiatry, 66(7), 642–665. https://doi.org/10.1177/0020764020933287
Schneider-Kamp, A., & Fersch, B. (2021). Detached co-involvement in interactional care: Transcending temporality and spatiality through mHealth in a social psychiatry out-patient setting. Social Science & Medicine, 285, 114297. https://doi.org/10.1016/j.socscimed.2021.114297

Figures 1-4: The pictures are more confusing than helpful. First, they do not represent healthcare professionals. Second, they introduce and highlight additional dimensions of gender, ethnicity etc. that I do not find in the description of the personas to a considerable degree.

Lines 496-500: Such suggestions would profit from being embedded in mobile app-based solutions for the care of patients with psychiatric disorders under the (remote) guidance of healthcare professionals. See the two articles on mHeath above, in particular, Schneider-Kamp & Fersch, which shows how healthcare professionals can get and stay involved in patients' lifes and everyday practices.

Lines 643-688: You should discuss possible issues stemming from the use of personas here. See the two critical articles mentioned above, in particular, Salminen et al., which mentions a list of potential issues also with manually-created personas from qualitative data.

Line 729: I assume "V" is for "F" and this is an artefact of not translating "vrouw" from Dutch to English?

Author Response

Dear reviewer,

We are thankful for the reviewers’ suggestions and believe that the manuscript was greatly improved by their suggestions.

Below you can find the detailed comments, our responses and changes in the manuscript (these are in red in the manuscript). 

Thank you for an interesting read. While the context of psychiatric disorders is not my main field of interest, I found the work highly relevant to the on-going debate on how to care for vulnerable patient groups. That said, the manuscript needs to be improved in two major aspects to be considered for publication.

Response

We thank the reviewer for the compliments.

Comment #1
First, the methodological approach of persona generation from qualitative data might or might not be appropriate. The manuscript currently only takes a positive view on personas, ignoring the widely spread critical voice that have surface in the wake of the popularity of personas in design. In the detailed comments below, I make some recommendations with which works to engage and where and how to improve the manuscript.

Response
Thank you for the relevant remarks. We took all your recommendations into account while writing and improving our manuscript.  With regard to the remarks about the used study design and its limitations, please see our answer on comment 4, comment 9 and comment 11.

Comment #2
Second, the discussion currently reads more like a summary of findings with a few managerial implications. The manuscript needs to be improved by engaging with more of the extant work on the social dimensions of care for psychiatric patients, in general, and the use of digital tools, in particular. The detailed comments suggest some works to engage with in order to improve this aspect of the manuscript.

Response

Thank you for your relevant remarks. With regard to the remarks about the digital tools in particular, please see our answer on comment 8 and comment 11.

Comment #3

Line 24: Maybe "one or more psychotic disorders"?

Response

Thank you for this comment. Our study aimed to gain insight into the attitudes, barriers and needs of MHNs regarding oral health in patients with a psychotic disorder. A patient can have more psychiatric disorders, but only one psychotic disorder. A psychotic disorder is a vulnerability disorder. Therefore, we made no changes in the manuscript.

See the literature from McGorry, P.D.; Purcell, R.; Hickie, I.B.; Yung, A.R.; Pantelis, C.; Jackson, H.J. Clinical Staging: A Heuristic Model for Psychiatry and Youth Mental Health. The Medical journal of Australia 2007, 187, doi:10.5694/j.1326-5377.2007.tb01335.x. which shows the different stages of a psychotic disorder.

Comment #4

Line 208: You only cite proponents of personas. There has been a lot of critique, which you should at least mention and position relative to:
Chapman, C. N., & Milham, R. P. (2006). The Personas’ New Clothes: Methodological and Practical Arguments against a Popular Method. Proceedings of the Human Factors and Ergonomics Society Annual Meeting, 50(5), 634–636. https://doi.org/10.1177/154193120605000503
Salminen, J., Jung, S.-G., & Jansen, B. J. (2021). Are data-driven personas considered harmful?: Diversifying user understandings with more than algorithms. Persona Studies, 7(1), 48–63. https://doi.org/10.3316/informit.352977339951659

Response

Thank you  for this suggestion.  We agree with the reviewer that there are some issues using personas. We added this in our manuscript (lines 220-228): In contrast of Grudin and Pruitt [27,40], Chapman et al., [41] has more skepticism about the persona method and discusses several methodological and practical limitations. For example, they point out the problem that it is hard to verify to what extent a persona can represents a larger (parts of) population. [41]. In this research, personas were a part of the data-synthesis activity since it serves designers better have a specific person to empathize with when designing new interventions [42]. In this research the personas have the function of an “empathic hand-over” [24], by giving a living insight into the MHNs through reading of the personas (e.g., by members of a design team).

In the discussion lines 715-722: From a strict scientific point of view, we recognize a possible issue with external validity [41]. However, persona creation may be considered more a data-synthesis activity, rather than a data-analysis activity. This means that – based on data- (design)researchers build personas to display relatable and authentic members of a target group; since it servers designers better have a specific person to empathize with when designing new interventions [42]. Thus, although personas should be based on accurate data, ultimately they serve a design process; they are not tools to meticulously represent a whole population in all its characteristics.

Comment #5

Line 217: It might be more appropriate to use the term "categories" or "overarching categories" instead of "large clusters".

Response

We have changed this in our manuscript, line 233: The data within the contextual interviews were organized within five larger categories.

Comment #6

Line 226-229: You should keep to the reference style with the square-bracketed numbers.

Response

We have adapted the reference style in line with the author instructions. Line 242-246

Comment #7

Line 234: What do you mean by "discussed until a consensus was reached"? Please elaborate.

Response

Thank you for this suggestion. We have described the consensus process in more detail in lines 250-251: Differences between both researchers while interpreting and coding the data were discussed until a consensus was reached.

Comment #8

Lines 357-362: Here, you mention suggestions of developing a mobile app that support oral health for patients with psychiatric disorders. You do, however never come back to this in the discussion. I would suggest to amend the discussion on the co-creation of interventions in Lines 685-688 by engaging with current literature on mobile app-based solutions for supporting patients with psychiatric disorders:
Chivilgina, O., Wangmo, T., Elger, B. S., Heinrich, T., & Jotterand, F. (2020). mHealth for schizophrenia spectrum disorders management: A systematic review. International Journal of Social Psychiatry, 66(7), 642–665. https://doi.org/10.1177/0020764020933287
Schneider-Kamp, A., & Fersch, B. (2021). Detached co-involvement in interactional care: Transcending temporality and spatiality through mHealth in a social psychiatry out-patient setting. Social Science & Medicine, 285, 114297. https://doi.org/10.1016/j.socscimed.2021.114297

Response

Thank you for your suggestion. We added this in the discussion of our manuscript (lines 629-624): The development of a digital oral health screening tool can fit in the needs of MHNs. A recent review shows that the use of mobile health (MHealth) is promising for patients with a psychotic disorders [57]. But considering this, it is important to discuss the impact of mobile technologies on the professional relationship between MHNs and patient with a psychotic disorder. Here, the findings of Schneider-Kamp and Fersch [52] demonstrate that, technological solutions such as mHealth can improve some care processes, also for mentally vulnerable groups, such as patient with a psychotic disorder, and show how MHNs can get and stay involved in patients' lives and everyday practices. Thus, the use of MHealth can support to reinvest in time savings into the improvement of the MHN-patient relation. Additionally, it is important to point out that for the positive effects of detached co-involvement to emerge and to avoid repercussions, a fine balance must be struck between face-to-face and MHealth [52].

Comment #9

Figures 1-4: The pictures are more confusing than helpful. First, they do not represent healthcare professionals. Second, they introduce and highlight additional dimensions of gender, ethnicity etc. that I do not find in the description of the personas to a considerable degree.

Response

Thank you for this comment. It was important to reflect the personas in the article because, as a handover, they were crucial to our research methodology. For that reason we included them, the pictures are part of the methodology of persona development. The personas ultimately, they serve a design process; they are not tools to meticulously represent a whole population in all its characteristics.

We added this in lines 225-229: Pictures and background information were added to help designers to create an elaborate and relatable image of this person. Therefore, personas have the function of an “empathic hand-over” [24] by giving a living insight into the MHNs through their reading of the personas. 

Additionally, the member check rounds did not reveal that people found the text-image combination inconsistent or illogical.

Comment #10

Lines 496-500: Such suggestions would profit from being embedded in mobile app-based solutions for the care of patients with psychiatric disorders under the (remote) guidance of healthcare professionals. See the two articles on mHeath above, in particular, Schneider-Kamp & Fersch, which shows how healthcare professionals can get and stay involved in patients' lifes and everyday practices.

Response

Thank you for this comment.  In the manuscript we added this sentence in the discussion, line 612-613: Patients and MHNs can profit from these short videos, by embedding these videos as micro-learning intervention in a mobile application [52]

Comment #11

Lines 643-688: You should discuss possible issues stemming from the use of personas here. See the two critical articles mentioned above, in particular, Salminen et al., which mentions a list of potential issues also with manually-created personas from qualitative data.

Response

Thank you for your comment. We have made changes in the limitations section of the manuscript (highlighted in red).

Lines 703-708: The wide variety of sources we used in this study, as well as the persona validation which provided a rigorous check on personas, increased the ecological validity. Additionally, Salminen et al. [61] discusses a lack of transparency as one of the challenges of creating personas, but we are confident that the rigorous described iterations show how the personas were generated and therefore decrease the lack of transparency. Our findings described the development and validation of four personas that can be used as an empathic handover and communicate with designers of management staff about the needs of MHNs regarding oral health. Based on data from a variety of sources, including the literature, contextual interviews, and semi-structured interviews, we identified and presented unique oral care perspectives, the main barriers, needs, suggestions for interventions, and site conditions, which were synthesized through four personas. Lines 714-715: These four personas were manually-created because there are no pictures of average or stereotypical MHNs available [60].

Lines 716-723: One related issue in the creation of personas is that there are no empirically validated guidelines (e.g., as to how many personas should be created) [60]. For this research, we created four personas, data saturation and member checking showed that this was enough to give an answer on the research question. Although we are confident that these profiles can be widely recognized, it is possible that we have missed profiles. However, these personas give an indication for the next step.

Comment #12

Line 729: I assume "V" is for "F" and this is an artefact of not translating "vrouw" from Dutch to English?

Response

Thank you for your comment. In line 800 (appendix B), we have changed this “V” into “F” in our manuscript.

Reviewer 2 Report

Congratulations for the excellent study developed. Very interesting.

However, there are some minor issues that I suggest that can be improved before publication:

- In the methods, the type of study is not clear.

- It would be interesting to have some more quantitative data presented. It might not be possible, but if so, one or two tables with some prevalences of specific type of answers could help the reader have a better perception of the answers given. The application of graphs could help also.

- The strategies proposed for the future exist in the manuscript. However, I believe that the description of specific guidelines in the conclusions or in the discussion section could be a benefit as future recommendations.

Author Response

Dear reviewer,

We are thankful for the reviewers’ suggestions and believe that the manuscript was greatly improved by their suggestions.

Below you can find the detailed comments, our responses and changes in the manuscript (in red). 

Congratulations for the excellent study developed. Very interesting. However, there are some minor issues that I suggest that can be improved before publication

Response

We thank the reviewer for the compliments. We took all your recommendations into account while writing and improving our manuscript.

Comment #1

In the methods, the type of study is not clear.

Response

We have described in more detail in lines 108-110: For this study, a qualitative, descriptive, interpretative design was conducted to gain insight into the attitudes, barriers and needs of MHNs regarding oral health in patients with a psychotic disorder. To determine the contributions to practice, a human-centered design (HCD) approach was adopted. An HCD approach is a co-creative, iterative, and creative approach with non-linear steps to problem-solving to provide tailor-made solutions for wicked problems [25].

Comment #2

It would be interesting to have some more quantitative data presented. It might not be possible, but if so, one or two tables with some prevalences of specific type of answers could help the reader have a better perception of the answers given. The application of graphs could help also.

Response

Thank you for this comment. We added three figures (appendix D, E, F) and a table (Appendix G) to give more insight in de frequencies of answers on elements of personas given by MHNs. We agree with the reviewer that this gives a better perception of the answers given by MHNs.  We added these on the last pages of this rebuttal.

Comment #3

The strategies proposed for the future exist in the manuscript. However, I believe that the description of specific guidelines in the conclusions or in the discussion section could be a benefit as future recommendations.

Response

Thank you for this comment. We added  this into the section conclusion and implication for practice (lines 772- 775):  Last, in this study, we described different strategies for MHNs to increase oral health in patients with a psychotic disorder. It is important to take these strategies into account while writing new guidelines (or editing existing guidelines) regarding oral health in patients with a psychotic disorder.

Round 2

Reviewer 1 Report

The authors have clarified all major issues and improved the manuscript to a publishable state.